# Anticipation-induced delta phase reset improves human olfactory perception

**Ghazaleh Arabkheradmand**[1], **Guangyu Zhou**[1]*, **Torben Noto**[1], **Qiaohan Yang**[1], **Stephan U. Schuele**[1], **Josef Parvizi**[2], **Jay A. Gottfried**[3,4], **Shasha Wu**[5], **Joshua M. Rosenow**[6], **Mohamad Z. Koubeissi**[7], **Gregory Lane**[1], **Christina Zelano**[1]

**1** Northwestern University Feinberg School of Medicine, Department of Neurology, Chicago, Illinois, United States of America, **2** Laboratory of Behavioral and Cognitive Neuroscience, Department of Neurology and Neurological Sciences, Stanford University Palo Alto, Stanford, California, United States of America, **3** University of Pennsylvania, Perelman School of Medicine, Department of Neurology, Philadelphia, Pennsylvania, United States of America, **4** University of Pennsylvania, School of Arts and Sciences, Department of Psychology, Philadelphia, Pennsylvania, United States of America, **5** University of Chicago, Department of Neurology, Chicago, Illinois, United States of America, **6** Northwestern University Feinberg School of Medicine, Department of Neurosurgery, Illinois, United States of America, **7** George Washington University, Department of Neurology, Washington DC, United States of America

* guangyu.zhou@northwestern.edu

**Data Availability Statement:** All source data are available at https://github.com/zelanolab/PhaseResettingInOlfactoryAnticipation.git.

## Abstract

Anticipating an odor improves detection and perception, yet the underlying neural mechanisms of olfactory anticipation are not well understood. In this study, we used human intracranial electroencephalography (iEEG) to show that anticipation resets the phase of delta oscillations in piriform cortex prior to odor arrival. Anticipatory phase reset correlates with ensuing odor-evoked theta power and improvements in perceptual accuracy. These effects were consistently present in each individual subject and were not driven by potential confounds of pre-inhale motor preparation or power changes. Together, these findings suggest that states of anticipation enhance olfactory perception through phase resetting of delta oscillations in piriform cortex.

## Introduction

Anticipation enhances sensory perception by improving stimulus detection and discrimination [1–8]. Anticipation-driven sensory enhancements can be dramatic in the human olfactory system, where detection thresholds of an anticipated smell may be significantly lower than thresholds for the spontaneous—that is, unanticipated—detection of the same odor. For example, industry standards of warning odorant concentration, such as the addition of ethyl-mercaptan to natural gas, can require a 57,000-fold increase in concentration over attended detection thresholds [9]. Yet the mechanism by which olfactory anticipation enhances perception is not well understood. In the visual and auditory systems, anticipation may impact perception through modulation of the excitability of local neuronal populations in sensory cortex [10–19] or through resetting of the phase of ongoing local field potential (LFP) oscillations prior to stimulus arrival [20–46]. Whether similar effects occur in the human olfactory system,

**Funding:** This work was funded by National Institutes of Health (NIDCD, https://www.nidcd.nih.gov/) grants R00DC012803 and R01DC016364 to C.Z. The funders had no role in study design, data collection and analysis, decision to publish, or preparation of the manuscript.

**Competing interests:** The authors have declared that no competing interests exist.

**Abbreviations:** BOLD, blood oxygen level dependent; CT, computed tomography; DMP, deviation from the mean phase; FDR, false discovery rate; fMRI, functional magnetic resonance imaging; ITPC, intertrial phase coherence; iEEG, intracranial electroencephalography; LFP, local field potential; MNI, Montreal Neurological Institute; MRI, magnetic resonance imaging; PSD, power spectral density.

with its phylogenetically older architecture and unique thalamic organization, is unknown. Though decades of research have focused on behavioral and neural correlates of attention in the visual and auditory systems [47–51], olfactory attention is far less studied.

Humans are capable of bringing selective attention to olfaction, which increases response speeds and impacts sniffing behavior [52–54]. Furthermore, olfactory attention modulates odor-evoked activity in the human brain [55–58], specifically in olfactory cortex [52,59–63]. Though fewer olfactory attention studies have focused on anticipation prior to the arrival of odor, enhanced functional magnetic resonance imaging (fMRI) signals have been observed prior to the onset of an odor in olfactory cortex, during anticipation [60,64]. However, fMRI measures of blood oxygen level dependent (BOLD) signals are unable to measure increases in excitability and phase synchrony within local neural populations. Thus, the oscillatory dynamics underlying olfactory anticipatory states are not well understood in the human brain. Olfactory attention also occurs in rodents, in whom it improves olfactory response times and thresholds [65–67]. A recent pioneering study showed that rats are capable of olfactory selective attention and that olfactory attentive and anticipatory states sharpen single-unit responses in the olfactory tubercle, resulting in increased stimulus contrast [66]. However, the importance of oscillatory phase dynamics—which have been shown to be critical during anticipation in other sensory systems—remain unclear in the olfactory system.

Numerous studies have established that LFP oscillatory dynamics are fundamental to olfactory processing mechanisms [68–83]. In rodents and humans, LFP oscillations in the olfactory bulb and in piriform cortex align to respiration [68,73,77,80,84–93], reflecting the fact that olfactory sampling is tied to breathing. Inhaling always brings with it the possibility of encountering an odor. In fact, the olfactory system allows fine control over the timing of an anticipated odor, through timing of the nasal inhale, which delivers odor molecules to receptors in the nose. Therefore, an investigation into the role of LFP oscillatory dynamics in olfactory anticipation will necessarily involve consideration of the relationship between those dynamics and the onset of inhalation, which is linked to the onset of odor.

We hypothesized that despite architectural and organizational differences between olfactory and other sensory systems [94], LFP phase would reset prior to inhalation onset—that is, stimulus onset—in the olfactory system, as is the case in other sensory systems. This would suggest that phase resetting as a neural signature of sensory anticipation is preserved across sensory systems of varied phylogenetic age and that the olfactory system is capable of similar mechanisms even without the precortical thalamic relay, which may be critical for these mechanisms in other modalities [1,30].

In this study, we used intracranial electroencephalography (iEEG) recordings from human piriform cortex (Fig 1A) to study the neural signatures of olfactory anticipation in the human olfactory system. Using iEEG, we were able to record LFPs with high temporal precision, allowing us to test the hypothesis that anticipation of odor would reset the phase of LFP oscillations in piriform cortex prior to inhale onset. We further hypothesized that if this phase reset was related to anticipation-induced enhancement of perception, then it would predict the strength and accuracy of subsequent neural and behavioral responses to the ensuing odor.

## Results

We investigated olfactory anticipation in a set of data gathered from neurosurgical patients with intracranial electrodes in piriform cortex who participated in olfactory tasks over the last 10 years, selected by inclusionary criteria (Fig 1B). This selection took advantage of the fact that most human olfactory tasks involve periodic, cued trials including presentation of odors, separated by fairly large intertrial intervals, during which participants breathe without

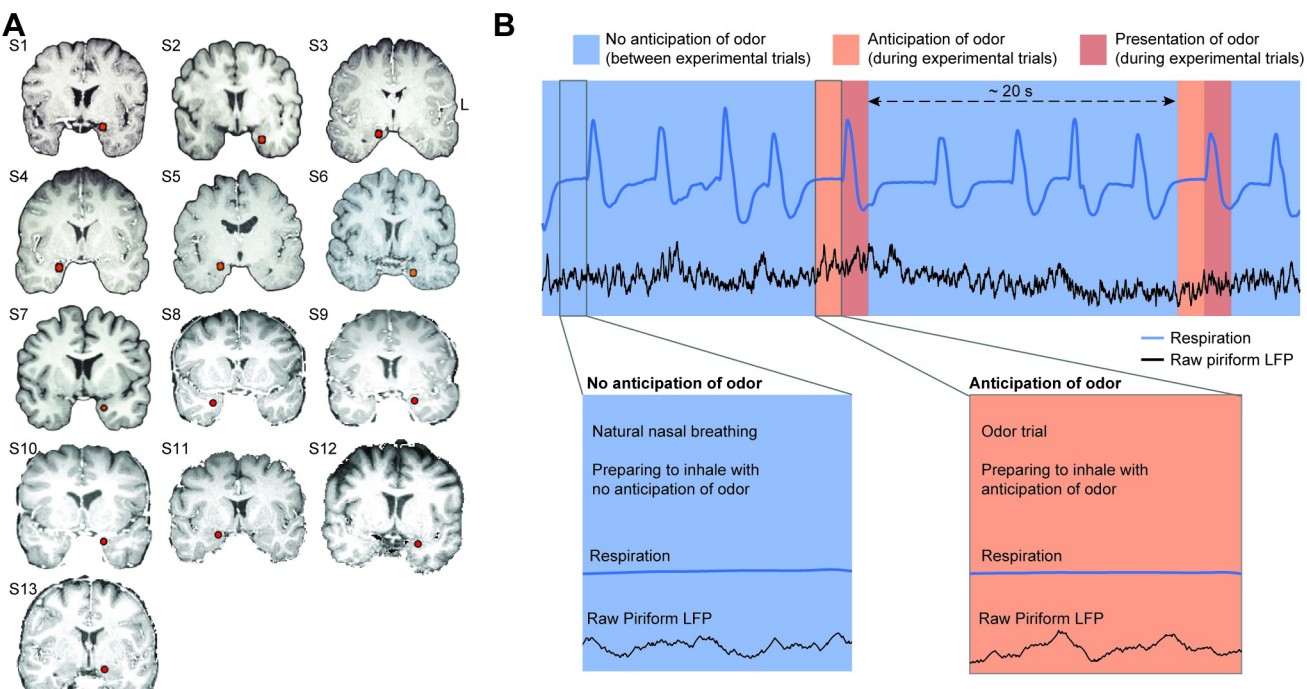

**Fig 1. Electrode locations and experimental conditions.** (A) The location of the piriform cortex electrode (red dot) is shown on each participant's brain image. L, left hemisphere. (B) Schematic illustration of anticipatory and nonanticipatory conditions. Inhales taken in anticipation of odor (during experimental trials) were compared to inhales taken without anticipation of odor (between experimental trials). Magnified panels show time windows of interest, which included the 1 s time window prior to inhale. Our analyses focused on the time just prior to inhale, when the subject was either anticipating or not anticipating an odor. LFP, local field potential.

anticipation of odor. Thus, data obtained from a typical olfactory task includes nasal inhalations taken in anticipation of an odor (experimental trials) and nasal inhalations taken without anticipation of odor (nasal inhalations taken between trials). We focused on the time windows just prior to these two types of inhalations, which constituted the two conditions of interest for this study—anticipatory and nonanticipatory. Inclusionary criteria were defined by the following considerations: In order to maintain a clear distinction between anticipatory and nonanticipatory inhalations, the time between trials must be adequately long; in order to test for the effects of anticipation on the ensuing olfactory coding, trials must include performance of a perceptual task; and finally, in order to precisely compare conditions, inhale onsets during both the trial and nontrial periods must be clear in the respiratory data. We therefore reviewed all of our olfactory iEEG data sets collected since 2010 in order to identify those which met these criteria: (1) human neurosurgery participants were periodically presented with cued odors, (2) the time between odor presentations exceeded 6 times the respiratory period of the individual, (3) participants performed an olfactory perceptive task, and (4) respiratory data were consistently of high enough quality to accurately determine inhale onsets both during and between trials. We identified data sets from 21 neurosurgical participants performing olfactory tasks, and, with our restrictions, 8 were excluded, leaving 13 participants who met our criteria and were included in the study (Fig 1A).

In each data set, we analyzed the time window just prior to nasal inhalations taken both with and without anticipation of odor (Fig 1B). In order to isolate the effects of anticipation from respiration or odor processing, we focused on a time period when subjects were engaged in a natural pause between breaths—neither inhaling nor exhaling—and on the verge of taking

their next breath, prior to any odor onset. Focusing on the pre-inhale time window ensured a highly controlled experimental comparison by allowing consideration of the effects of anticipation in the absence of sniffing behavior and in the absence of odor—two parameters which could potentially confound interpretation of results. Therefore, the only difference between conditions was whether the ensuing inhale would be taken with anticipation of an odor or without anticipation of an odor (Fig 1B).

## Anticipation resets the phase of human piriform oscillations

When attention is brought to odor—for example, when opening the lid of a cooking pot to smell its contents—humans enter into a deliberate state of anticipation just prior to nasal inhalation. Here, we tested the hypothesis that anticipation of odor resets the phase of LFP oscillations in human piriform cortex just prior to anticipatory inhalation. Previous studies in the visual and auditory systems have found a particular importance of delta range oscillations (0–3 Hz) in anticipatory mechanisms in humans and monkeys [21,32,35,39,95]. Therefore, we expected to find anticipatory phase resetting in the human olfactory system mainly at low frequencies. To test this hypothesis, we compared the phase coherence of oscillations prior to inhale onset across anticipatory trials (anticipatory condition) with phase coherence across nonanticipatory trials (nonanticipatory condition). We used Intertrial Phase Coherence (ITPC), which measures the consistency of instantaneous phase values over event-locked trials [96–98] to estimate phase resetting [32,98]. We first created inhale-onset-aligned epochs of anticipatory and nonanticipatory trials for each participant. We then computed the ITPC across a broad range of frequencies (0.5–200 Hz) and times (1 s prior to inhale onset to 3 s following). We found that just prior to anticipatory inhales, statistically significant ITPC was evident in the low-frequency range (0.5–1.5 Hz; Fig 2A and 2B, right). Just prior to nonanticipatory inhales, no ITPC was evident (Fig 2A and 2B, left). In a direct statistical comparison between the two, anticipatory ITPC was significantly stronger than nonanticipatory ITPC in the low-frequency range. Importantly, this effect was evident in both a combined analysis, including data from all trials across all participants (permutation test, false discovery rate [FDR] corrected $P < 0.05$; max Z = 4.38), and at the individual level, with significant increases evident in each participant (two-tailed paired $t$ test, $T_{12} = 3.62$, $P = 0.0035$; Figs 2C and 3A). The effect was specific to piriform cortex, with reduced ITPC values in electrodes located outside of piriform cortex on the same implanted depth wire (two-tailed paired $t$ test, $T_{12} = 2.37$, $P = 0.035$; Fig 2E). Though statistically significant ITPC was also evident at higher frequencies, there were no significant differences between conditions; this suggests that while higher frequency oscillations (across a broad range of beta and gamma bands) do align with inhalation, the phase clustering at these frequencies may not be modulated by anticipation.

In the combined analysis, it was evident that the increase in low-frequency phase coherence preceding anticipatory inhales began approximately 0.5 s prior to inhale onset (Fig 2A and 2B). To determine whether the timing of ITPC was consistent across participants, we divided the pre-inhale time window into two separate intervals: [−1 s, −0.5 s] and [−0.5 s, 0 s] prior to inhale onset. We then computed maximal low-frequency (0.5–2 Hz) ITPC values within each time interval for each participant separately. In a two-way repeated ANOVA computed with "anticipatory-state" and "time-window" as factors, we found that, overall, ITPC was significantly stronger in the anticipatory compared to nonanticipatory condition (main effect of anticipatory-state: $F_{1,12} = 12.53$, $P = 0.0041$; Fig 2D). This difference was statistically more significant in the later time interval, beginning 0.5 s prior to inhale onset, evidenced by a significant interaction between anticipatory-state and time-window factors ($F_{1,12} = 13.56$, $P = 0.0031$). This effect was further evidenced by a direct comparison of individual

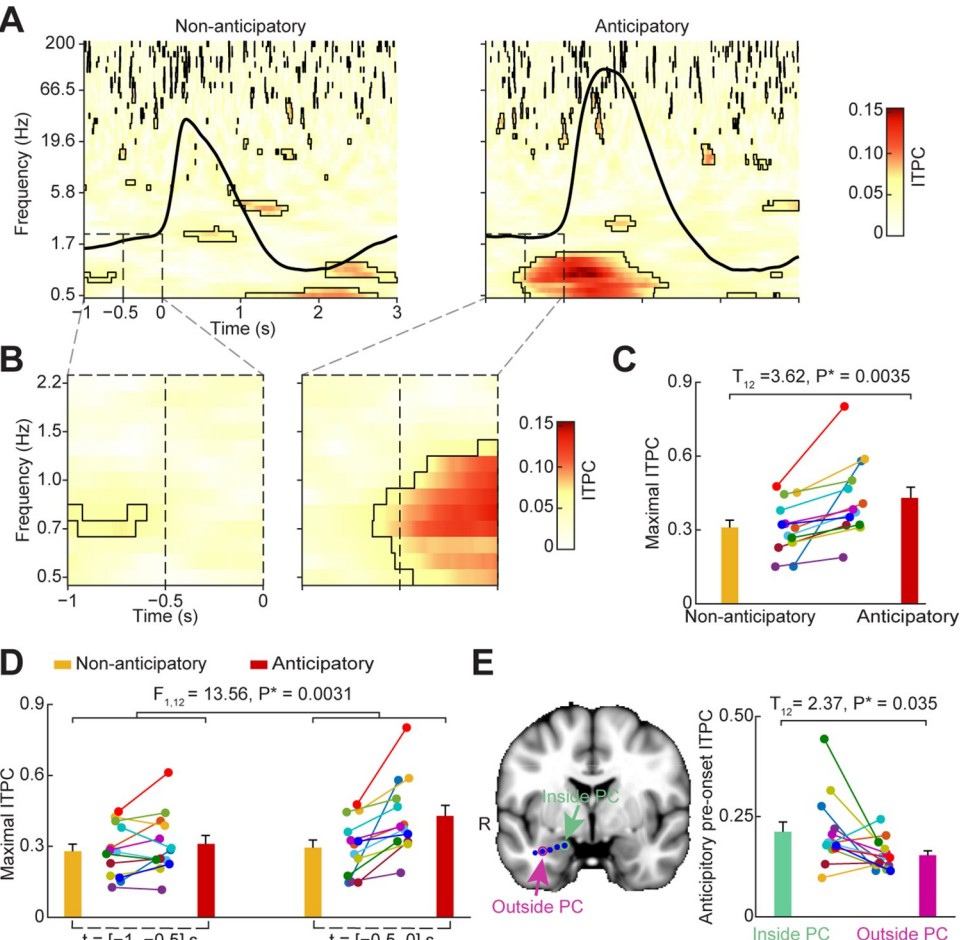

**Fig 2. Anticipation of odor induces phase reset prior to inhale.** (A) Time-frequency plot of phase reset. The t = 0 s indicates inhale onset. Black outlines indicate statistically significant clusters (FDR corrected $P < 0.05$). The subject-averaged respiratory signal is overlaid (black line). (B) Enlarged versions of the dashed-lined windows in panel A. (C) Individual maximal ITPC values in panel B. ITPC was higher in the anticipatory condition compared to the nonanticipatory condition (two-tailed paired *t* test; $T_{12} = 3.62$, $P = 0.0035$). (D) Individual maximal ITPC values across each half window in panel B. Peak ITPC values are plotted for each participant, for each time window, for each condition. An ANOVA analysis found a significant main effect of "anticipatory-state" ($F_{1,12} = 12.53$, $P = 0.0041$) and a significant interaction between "time-window" and "anticipatory-state" ($F_{1,12} = 13.56$, $P = 0.0031$). ITPC was significantly larger during the anticipatory condition compared to the nonanticipatory condition in the later time interval ($T_{12} = 4.09$, $P = 0.0015$) but not in the earlier time interval ($T_{12} = 1.56$, $P = 0.15$). (E) Phase reset was specific to piriform cortex. The ITPC was higher for electrodes located inside piriform cortex compared to those located outside of piriform cortex but on the same depth wire (two-tailed paired *t* test; $T_{12} = 2.37$, $P = 0.035$). The brain image on the left is a representative participant. The bar plot on the right shows the average ITPC for each subject in the time window from −0.5 s to 0 s pre-inhale onset and the frequency range from 0.5 Hz to 2 Hz. PC, piriform cortex; R, right hemisphere. Each colored dot represents one participant in panels C–E. The source data are available at https://github.com/zelanolab/PhaseResettingInOlfactoryAnticipation.git. FDR, false discovery rate; ITPC, intertrial phase coherence.

participants' maximal ITPC values within the anticipatory condition only, occurring within the two distinct time intervals (Fig 2D). In a two-tailed paired *t* test with multiple comparisons corrected using the Bonferroni method, we found that the maximal ITPC was significantly stronger in the time interval from [−0.5 s, 0 s], indicating that, in the anticipatory condition only, ITPC became significantly stronger 0.5 s prior to inhale onset (two-tailed paired *t* test, $T_{12} = 5.2$, $P = 0.00022$). Notably, in contrast, there was no significant ITPC difference between

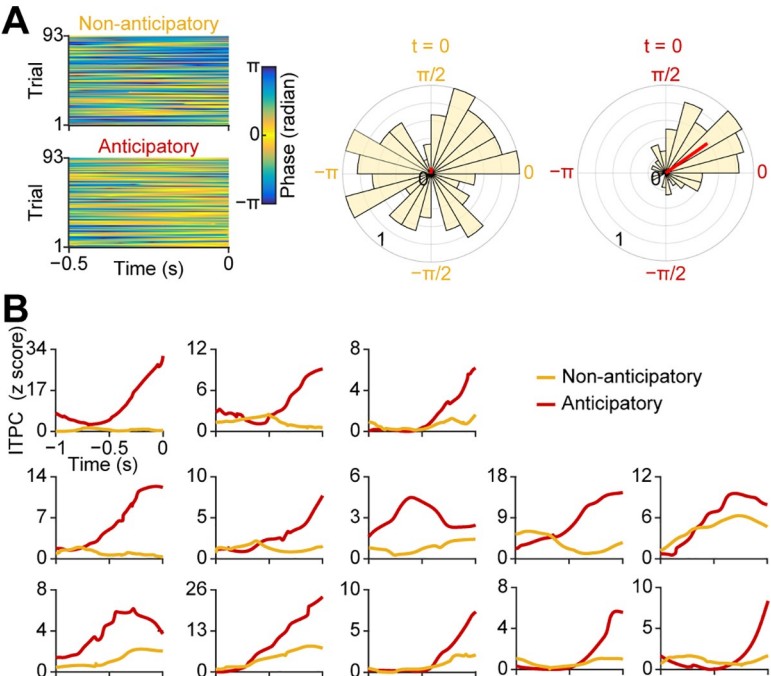

**Fig 3. Individual anticipatory phase resets prior to inhale onset.** (A) Single-trial instantaneous phase time series (0–2 Hz) for each condition from a representative participant. The raw instantaneous phase over time of the half second prior to inhale (−0.5 s to 0 s) is plotted onto the color map. Each row is a single trial. Increased ITPC is evident in the anticipatory condition as a more skewed color distribution (in the "yellow direction") compared to the nonanticipatory condition. Corresponding rose plots show the normalized distribution of single-trial phases at the inhale onset (t = 0 s) for each condition. The length of the red line on the rose plots indicates the magnitude of the ITPC for each condition. (B) Time course of phase resetting in the pre-inhale time window. For each participant, the Rayleigh z-score of the ITPC (at the maximal frequency from 0.5–2 Hz for each individual) time series is plotted over the 1 s pre-onset time window for each condition. The source data can be found at https://github.com/zelanolab/PhaseResettingInOlfactoryAnticipation.git. ITPC, intertrial phase coherence.

the two time intervals for the nonanticipatory condition (two-tailed paired $t$ test, $T_{12} = 0.83$, $P = 0.42$). We also computed the averaged time series of ITPC over the entire pre-inhale time window for each individual participant. The increase in ITPC was evident beginning around 0.5 s prior to inhale onset in each individual (Fig 3A and 3B). Taken together, these findings indicate that anticipation resets the phase of ongoing neuronal oscillations in human piriform cortex just before stimulus arrival.

## Anticipation does not modulate the power of human piriform oscillations

Phase and power are mathematically independent variables, but ITPC measures can be sensitive to power modulations, which can impact the precision of phase estimates [98,99]. Therefore, in order to test whether differences in power between experimental conditions affected measured differences in ITPC between experimental conditions in our data, we conducted a series of analyses designed to quantify relationships between power and ITPC during the pre-inhale time window. First, we determined whether there were any gross differences in pre-inhale power spectral density (PSD) across conditions. We computed the PSD (from 0–200 Hz, using Matlab's *pwelch.m* function) for each participant for each condition during the pre-inhale time window. We found no significant differences in PSD across conditions during the pre-inhale time window (FDR corrected T-statistics, $P > 0.05$; Fig 4A). Of note, in human piriform cortex we did not see an obvious PSD theta peak, which has been consistently found in

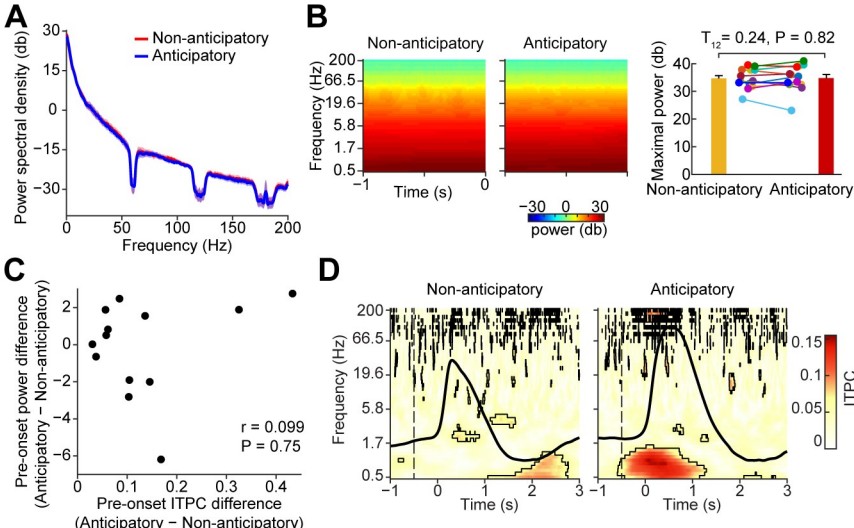

**Fig 4. Anticipatory phase reset is not related to pre-inhale power.** (A) PSD of the pre-inhale time window. The average PSD is plotted for each condition. The shaded area indicates standard error over participants ($N = 13$). PSD did not differ across conditions (FDR corrected $P > 0.05$). (B) Pre-inhale time-frequency analysis. Average raw power is plotted as a color map for both conditions (two left panels). The bar plot on the far right shows anticipatory peak power (0.5 Hz–12 Hz, from −1 s to 0 s prior to inhale onset) for both conditions. There was no significant difference across conditions (two-tailed paired $t$ test; $T_{12} = 0.24$, $P = 0.82$). Each colored dot represents data from one participant. (C) Pre-inhale phase reset and pre-inhale power are not correlated. The maximal pre-inhale ITPC and power were calculated over 0.5 Hz to 2 Hz and −0.5 s to 0 s prior to inhale onset. The maximal ITPC difference between anticipatory and nonanticipatory conditions is not correlated to power (Spearman correlation, r = 0.099; $P = 0.75$). (D) Related to Fig 2A, time-frequency plot of phase reset using a shorter filter window. The t = 0 s indicates inhale onset. Black outlines indicate statistically significant clusters (FDR corrected $P < 0.05$). The subject-averaged respiratory signal is overlaid (black line). The source data are available at https://github.com/zelanolab/PhaseResettingInOlfactoryAnticipation.git. FDR, false discovery rate; ITPC, intertrial phase coherence; PSD, power spectral density.

other medial temporal lobe structures such as the hippocampus [100]. Second, because PSD calculations cannot detect power changes over time, we tested whether any condition-related differences in LFP power arose over the course of the pre-inhale time window. For each individual participant, we conducted a Hilbert-based time-frequency analysis of the instantaneous power over frequencies from 0.5 Hz to 200 Hz during the 1 s time window prior to inhale onset. In this analysis, baseline correction can impact signal-to-noise ratio, which impacts ITPC [99]; therefore, we conducted this analysis using raw data without baseline correction. We found no significant differences in power across conditions during the pre-inhale time window (FDR corrected $P > 0.05$, Fig 4B). This was true both in a combined analysis, including trials across all subjects (Fig 4B, left), and in an individual analysis in each subject (Fig 4B, right, bar plot; two-tailed paired $t$ test; $T_{12} = 0.24$, $P = 0.82$). Third, even though we found no differences in LFP power across conditions using the PSD or Hilbert measures, it was still possible that small changes in power across conditions could impact differences in the strength of ITPC across subjects. We therefore looked for a relationship between pre-inhale ITPC and differences in pre-inhale power across conditions across subjects. We found no relationship between these measures in either the 0.5 Hz to 2 Hz or 0.5 Hz to 8 Hz frequency ranges (Fig 4C; frequency range of 0.5–2 Hz: Spearman correlation, r = 0.0.099, $P = 0.75$; Pearson correlation, r = 0.22, $P = 0.48$; frequency range of 0.5–8 Hz: Spearman correlation, r = 0.27, $P = 0.37$; Pearson correlation, r = 0.25, $P = 0.42$). Finally, it is possible that "temporal leakage" due to inherent temporal smoothing during filtering could have caused poststimulus effects to impact

measures in the prestimulus time window. To control for this possibility, we reran our time-frequency analysis using a shorter filter window (2 cycles per frequency) [101]. Virtually identical results were found using the shorter window length (Fig 4D), confirming that our findings were not due to filter smearing effects.

Combined, these findings suggest that increases in ITPC during odor anticipation are independent of differences in power, and therefore reflect true phase resetting.

## Anticipatory phase reset correlates with odor-evoked responses

Anticipation enhances perception. We hypothesized that if phase resetting is a neural signature of anticipation, then it should relate to odor coding and perceptual accuracy. Therefore, we next looked for relationships between pre-inhale ITPC and measures—both neural and behavioral—of olfactory perception following odor presentation.

Odor-evoked theta-band responses in human piriform cortex may carry odor-identity information [102] and appear to play an important role in odor coding in rodents [68,103,104]. Based on these findings, we reasoned that if anticipatory phase resetting is related to odor coding, then it should predict ensuing odor-induced theta power. To test this hypothesis, we looked for a relationship between the pre-inhale ITPC and the postinhale increase in power between conditions. We first established that odor-induced theta power following inhale onset was stronger in the anticipatory compared to the nonanticipatory condition (Fig 5A and 5B). We found that theta power was significantly increased during anticipatory trials compared to nonanticipatory trials in both a combined analysis and in each individual participant (combined analysis: FDR corrected $P < 0.05$; individual analysis: two-tailed paired $t$ test; $T_{12} = 5.68$, $P = 0.000102$). We then asked whether odor-evoked theta power increases were related to anticipatory phase resetting. To do this, we looked for a predictive relationship between the pre-inhale ITPC and the postinhale theta power increase across subjects. We found a robust positive correlation between the strength of ITPC during anticipation, and odor-induced theta power increases across subjects (Spearman correlation, r = 0.76, $P = 0.0036$; Fig 5C). This finding suggests that phase resetting of ongoing LFP oscillations prior to the arrival of anticipated odor impacts subsequent processing of that odor. Notably, the observed odor-evoked increase in theta power in the olfactory anticipation condition could be explained by the consistent delta phase reset and phase amplitude coupling. Thus, although a clear relationship was found between anticipatory signatures and subsequent odor processing, the oscillatory mechanism underlying this subsequent odor processing cannot be definitively determined by our data.

## Anticipation-induced phase reset improves olfactory perceptual accuracy

If anticipation matters for odor perception, then it should impact behavioral outcomes. Therefore, we conducted two separate analyses that were each designed to look for a relationship between pre-inhale phase reset and perceptual accuracy on the ensuing task. To do this, given the relatively small number of incorrect trials per subject, and the fact that ITPC measures stabilize at around 20 to 30 trials [98], we used a single-trial measure of ITPC: the deviation from the mean phase (DMP) measure [105,106]. DMP provides an estimate of how well each trial aligns to the clustered phase angle, thus constituting a single-trial measure of phase resetting. In this analysis, for a given trial, a small DMP value would indicate the presence of phase reset, whereas a large DMP value would indicate a lack of phase reset. First, for each participant, we grouped trials according to their perceptual accuracy (correct or incorrect) and then examined the DMP for each group. Second, for each participant, we grouped trials according to their DMP (well-aligned versus poorly aligned) and then examined the perceptual accuracy for each

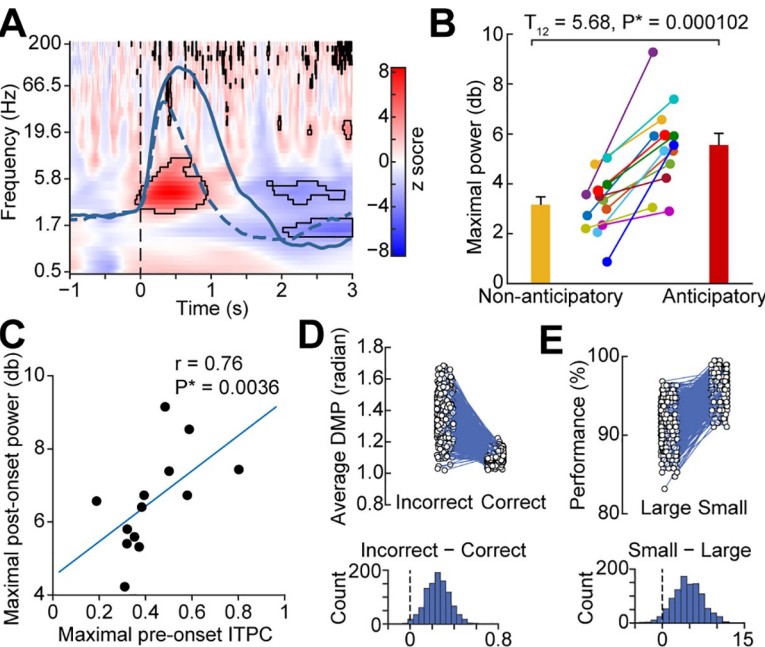

**Fig 5. Anticipatory phase reset correlates with odor-induced theta power and olfactory perceptual accuracy.** (A) Anticipation prior to inhale increases theta power following inhale. The difference in power across conditions is shown. Black outlines indicate statistically significant differences (permutation test, FDR corrected $P < 0.05$). The vertical dotted line indicates inhale onset, and solid and dashed overlays indicate subject-averaged respiratory signals for anticipatory and nonanticipatory conditions respectively. (B) Anticipation-induced theta power increases are evident in each individual participant. The maximal power over the frequency range 0.5 Hz to 12 Hz and 0 s to 1 s following inhale onset is plotted for each participant. Anticipatory power is greater than nonanticipatory power across participants (two-tailed paired $t$ test, $T_{12} = 5.68$, $P = 0.000102$). Each colored dot represents data from one participant. (C) Anticipatory phase reset prior to inhale onset correlates with odor-evoked theta power across subjects. The maximal ITPC from $-0.5$ s to 0 s pre-inhale and 0.5 Hz to 2 Hz for each participant is plotted against the maximal theta power following inhale onset for each participant. A significant correlation was evident (Spearman correlation, $r = 0.76$, $P = 0.0036$). Each dot indicates one participant. (D) Single-trial phase resetting prior to inhale onset predicts olfactory perceptual accuracy. The DMP is plotted for correct and incorrect trials computed on each repetition ($N = 1,000$) of a bootstrap analysis (upper panel). Each dot is the DMP value for each condition for that repetition, with lines connecting the values obtained on each repetition. The distribution of the difference is plotted (lower panel), which was statistically compared to zero. Correct trials were more closely aligned to the mean phase than incorrect trials, indicating stronger phase resetting prior to inhale on trials when the participant later made an accurate olfactory perceptual decision (percentage-based $P = 0.009$; sign-test $P = 2.72 \times 10^{-211}$). (E) Single-trial phase resetting predicts olfactory perceptual accuracy. The behavioral accuracy is plotted for trials with small and large pre-inhale DMP values, computed on each repetition ($N = 1,000$) of a bootstrap analysis (upper panel). Each dot is the accuracy for each condition for that repetition, with lines connecting the values obtained on each repetition. The distribution of the difference is plotted (lower panel), which was statistically compared to zero. Trials with small DMP values resulted in higher accuracy on the task than those with large DMP values, indicating that stronger phase resetting occurred prior to inhale on trials when the participant later made an accurate olfactory perceptual decision (percentage-based $P = 0.042$; sign-test $P = 3.66 \times 10^{-191}$). The source data are available at https://github.com/zelanolab/PhaseResettingInOlfactoryAnticipation.git. DMP, deviation from the mean phase; FDR, false discovery rate; ITPC, intertrial phase coherence.

group. For both analyses, bootstrapped distributions of difference values were computed across conditions, the mean of which was subsequently compared to zero (Fig 5D and 5E). In our first analysis, we found that correct trials had smaller pre-inhale DMP values compared to incorrect trials (making no assumption of a normal distribution, 991 out of 1,000 repetitions yielded a difference $> 0$, $P = 0.009$; if we assume a normal distribution, $Z = 2.3$, $P = 0.01$; paired $t$ test, $T_{999} = 72.58$ $P = 0$; sign-test $P = 2.72 \times 10^{-211}$), suggesting better alignment with the mean phase for correct over incorrect trials (Fig 5D). In our second analysis, we found that trials with small pre-inhale DMP had better ensuing perceptual accuracy compared to trials with

large pre-inhale DMP (Fig 5E; making no assumption of a normal distribution, 958 out of 1,000 repetitions yielded a difference $> 0$, $P = 0.04$; if we assume a normal distribution, $Z = 1.8$, $P = 0.03$; paired $t$ test, $T_{999} = 57.31$, $P = 0$; sign-test $P = 3.66 \times 10^{-191}$). Thus, results of these two complementary analyses suggest that anticipatory phase resetting in piriform cortex improves perceptual accuracy.

### Anticipatory phase reset does not correlate with olfactory motor behavior

Anticipation induces activation of motor and premotor areas prior to arrival of a stimulus, likely in preparation for movement relevant to the expected stimulus [107–111]. In line with this, anticipation of odor has been shown to impact both olfactory perception (behavior and coding) [64,66,112,113] and olfactory sampling (sniffing/motor) behavior [52,114]. It is therefore possible that anticipatory phase resetting in piriform cortex relates to both olfactory perception and olfactory motor behavior. In the previous section, we established a link between anticipatory pre-inhale phase resetting and olfactory perceptual responses. To determine whether this link was specific to perceptual effects of anticipation and not confounded by olfactory motor responses, we next asked whether there was also a link between anticipatory phase resets and subsequent olfactory motor behavior. To do this, we looked for a predictive relationship between prestimulus ITPC and the ensuing inhale peak and duration, both across and within subjects.

First, we examined the respiratory signal of each subject following inhale onset for anticipatory and nonanticipatory conditions separately (Fig 6A). Confirming findings from previous studies, we found that inhale peaks and durations were significantly larger following inhale onset for the anticipatory compared to the nonanticipatory condition (Peak value: two-tailed paired $t$ test, $T_{12} = 9.04$, $P = 1.05 \times 10^{-6}$; Fig 6B, left. Inhale duration: two-tailed paired $t$ test, $T_{12} = 5.34$, $P = 0.00018$; Fig 6B, right), suggesting a link between olfactory anticipation and olfactory motor behavior. However, it does not necessarily follow from this finding that there is a link between anticipatory phase resetting and olfactory motor behavior. To address this possibility, we asked whether the strength of the pre-inhale ITPC predicted the size of the ensuing inhale; if true, this would suggest a link between pre-inhale phase resetting and motor-related anticipatory correlates. We computed the difference in ITPC between anticipatory and nonanticipatory conditions and compared this to the difference in inhale peak value between the two conditions across participants. We found no statistically significant correlation between pre-inhale ITPC increase prior to inhale onset and ensuing inhale peak (Spearman correlation, r = 0.15, $P = 0.63$; Pearson correlation, r = 0.4, $P = 0.15$) or inhale duration (Spearman correlation, r = 0.011, $P = 0.98$; Pearson correlation, r = −0.24, $P = 0.44$; Fig 6C). However, the lack of a significant correlation across subjects did not rule out the possibility of effects within subjects. To confirm the across-subject findings at a within-subjects level, we next used a median split of the DMP to separate each subject's trials into groups of well-aligned (small DMP) and poorly aligned (large DMP) phase at onset. We then compared the inhale sizes for the trials from each group. We found no relationship between the degree of phase alignment at inhale onset and the size of the ensuing inhale at the individual level (Fig 6D). These findings suggest that increased phase resetting during anticipation of odor does not impact respiratory behavior and is therefore unlikely to relate to the motor component of olfactory anticipation.

## Discussion

In this study, we found that anticipation of odor resets the phase of low-frequency LFP oscillations in piriform cortex prior to inhale onset. The strength of this phase reset was positively

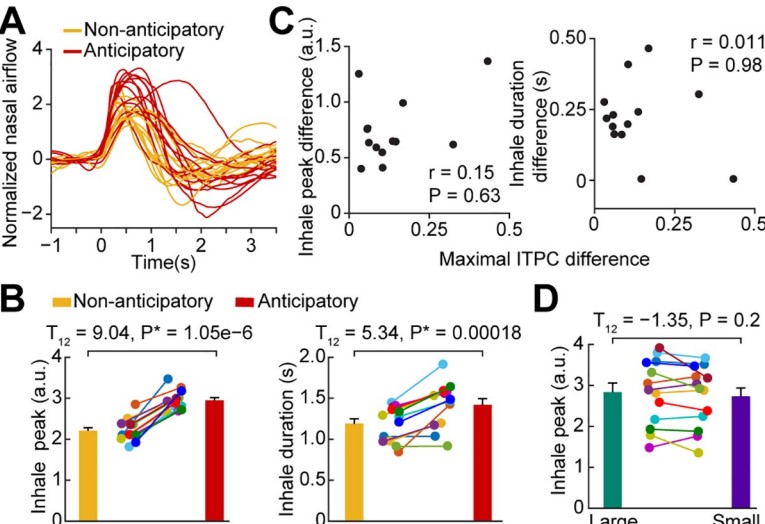

**Fig 6. Anticipatory phase reset does not relate to subsequent olfactory motor behavior.** (A) Average respiratory signals for each participant; anticipatory (red) and nonanticipatory (yellow) conditions. (B) Inhale peaks and durations. Each colored dot represents data from one participant. The anticipatory condition had both higher inhale peaks (left, $T_{12} = 9.04$, $P = 1.05 \times 10^{-6}$) and longer durations (right, $T_{12} = 5.34$, $P = 0.00018$) compared to nonanticipatory condition. Statistics were performed using a two-tailed paired $t$ test. (C) Anticipatory phase resetting is not correlated to changes in olfactory motor behavior across subjects. Each colored dot represents data from one participant. No significant correlation was found between pre-inhale ITPC and subsequent inhale peaks (Spearman correlation, r = 0.15, $P = 0.63$; Pearson correlation, r = 0.42, $P = 0.15$) or durations (Spearman correlation, r = 0.011, $P = 0.98$; Pearson correlation, r = −0.24, $P = 0.44$). (D) Anticipatory phase resetting is not correlated to changes in olfactory motor behavior within subjects. For each subject, we computed inhale peaks for trials median split by small and large DMP values. We found that trials with poor phase alignment had inhale peaks similar to those in trials with good phase alignment (two-tailed paired $t$ test, $T_{12} = -1.35$, $P = 0.2$). Each colored dot represents data from one participant. The source data are available at https://github.com/zelanolab/PhaseResettingInOlfactoryAnticipation.git. a.u., arbitrary units; DMP, deviation from the mean phase; ITPC, intertrial phase coherence.

correlated with anticipation-induced increases in odor response magnitude but not with anticipation-induced increases in inhale size. We also found that anticipatory phase reset improves olfactory perceptual accuracy. Together, these findings suggest that anticipation may enhance olfactory perception through neural dynamics that result in resetting of LFP phase to the timing of stimulus arrival. Notably, we did not observe changes in oscillatory power during olfactory anticipation, highlighting the importance of oscillatory phase dynamics over power modulations in olfactory anticipation. Because LFP oscillations represent rhythmic increases and decreases in neuronal excitability [30,115], our results suggest that olfactory anticipation may bring LFP oscillations to an optimal phase at the time of odor arrival (in this case, the onset of nasal inhalation). This study was limited to a focus on the oscillatory dynamics of low-frequency (delta) oscillations in piriform cortex. Future studies are needed to characterize the impacts of anticipatory—or nonanticipatory—states on higher frequency ranges.

Compared to other sensory systems, neural signatures of olfactory anticipation are less understood. In the visual and auditory systems, it is well established that phase reset of ongoing LFPs occurs during anticipation, which is linked to improved perception [20–36,40–46]. Here, we established that anticipatory phase reset occurs in the olfactory system, despite its distinct paleocortical architecture and its lack of a precortical thalamic relay. Our study extends findings from previous fMRI studies on human olfactory attention showing that BOLD activity in piriform cortex is modulated by attention to odor, during both anticipatory and poststimulus attentional periods [52,59–61]. Because fMRI is not a direct measure of neural activity, our finds suggest that the observed anticipatory phase shift may be detectable (indirectly) by fMRI

techniques. Furthermore, multivariate analysis methods have shown that spatial patterns of activity in piriform cortex during odor anticipation are stimulus-specific, representing the particular odor being anticipated [64]. However, a limitation of BOLD signal is its poor temporal resolution, which precludes analysis of oscillatory and phase dynamics of LFP oscillations. Our use of iEEG methods overcame these limitations, allowing us to measure LFP oscillations directly from human piriform cortex and to show that oscillatory phase, but not power, is modulated by anticipation of odor. Although iEEG methods have excellent temporal precision, they have limited spatial coverage; thus, future studies using both fMRI and iEEG techniques in the same participant pool would provide a powerful tool for measurement of both LFPs and multivariate spatial patterns and allow for a thorough investigation of the spatial extent of anticipatory effects across the many regions of primary olfactory cortex. For example, a recent study in rodents found olfactory attentional effects in the olfactory tubercle [116], from which we were not able to record in this study. Furthermore, a recent study reported modulations of the timing of transitions between metastable states during gustatory expectation [117], indicating that future work on metastability in human olfactory cortex is needed.

In interpreting our results, there is potential for confusion over the words "expectation" and "anticipation," which are sometimes used interchangeably. Here, we use the terms "anticipation" and "anticipatory" to mean the general belief, prior to inhalation, that odor will be present on the next inhale, without expectations about specific identity or characteristics of the odor and without regard to any reward/punishment. Referring to "expectation(s)" about a coming odor may imply that the subject holds a belief or set of beliefs about characteristics or context of the coming stimulus (for example, labeling of the same chemical as either "cheese" or "vomit" influences olfactory perception of that chemical). Expectations about a coming odor have been shown to impact perception [118–124]. This is distinct from general anticipation of an odor, without beliefs about the characteristics of that odor. In addition, the study of reward anticipation following a stimulus [67,125–127] is also distinct from our use of "anticipation," here, where we consider general anticipation of odor.

Our findings dovetail with current theories of predictive coding, which suggest that top-down influences convey predictive signals at lower frequencies compared to bottom-up signals, which convey prediction errors at higher frequencies [128–133]. We found that the phase of low-frequency oscillations was reset during anticipation, prior to stimulus arrival. Though predictive coding studies have found top-down signals represented in beta-range oscillations [134–137], our results, in keeping with other studies [21,32,35,39,95], found anticipatory phase resetting in delta range oscillations. Previous studies have shown that prediction-related neural signals carry feature-specific information across modalities [64,138–142]. In our study, participants were told that an odor was coming, but they did not receive information that could be used to predict its identity. Thus, our study examined the state of general olfactory anticipation, analogous to general anticipation of tastes that have been studied in rodent gustatory cortex [6,117]. While predictive coding may involve multivariate spatial patterns of activity prior to the arrival of a stimulus [64,143–145], our data suggest these patterns may be combined with shifts in the phase of ongoing LFPs. Additional studies on olfactory anticipation will clarify these remaining questions.

A prominent idea is that sensory cortex oscillations entrain with rhythmic stimuli, especially during sustained attention [42–44,146]. When stimuli are not rhythmic, or during inattentive states, cortical oscillations are decoupled from stimuli and are characterized by different oscillatory properties. However, how this idea would apply to the olfactory system, where the stimuli are odor chemicals which lack inherent temporal structure, is unclear [147]. Here, we show that in human olfaction, low-frequency oscillations align with the timing of

stimulus onset by synchronizing with inhalation. In this way, anticipatory mechanisms in the olfactory system may involve oscillatory synchrony between respiration—which controls the timing of stimulus arrival—and cortex, analogous to the mechanism underlying entrainment between rhythmic stimuli and cortex in vision and audition. This suggests that there is a yet-to-be-discovered olfactory "unattended state," even though the olfactory system is, notably, never decoupled from respiratory rhythms [89]. We hypothesize that olfactory unattended states involve changes in higher frequency oscillations, as observed in vision and audition [147–149].

## Methods

### Ethics statement

The study was approved by the Institutional Review Boards of Northwestern University, University of Chicago, and Stanford University, and the study adhered to the declaration of Helsinki. Written informed consent was obtained from all participants.

### Participants

Our study included iEEG data sets from 13 participants with medication-resistant epilepsy. All participants had depth electrodes implanted stereotactically for clinical presurgical evaluation (Table 1 and Fig 1A). Electrode locations were determined solely based on clinical need and, in our participant population, included piriform cortex and amygdala within the medial temporal lobe. Data were acquired at Northwestern Memorial Hospital (10 participants), University of Chicago (2 participants), and Stanford University (1 participant).

Subjects participated in olfactory tasks that involved periodic presentation of cued odors with time periods between odor trials exceeding 6 times the respiratory period of the individual. During this intertrial period, subjects breathed naturally through the nose and no odor was anticipated or presented. In each case, the task included an olfactory perceptual behavioral component. Each data set included respiratory data that was of consistent high quality both during and between trials. The experimental tasks were conducted in each participant's hospital room and were computer-controlled, presented to subjects using an Apple laptop computer running Matlab (RRID: SCR_001622) via the PsychToolBox extension (RRID: SCR_002881). Synchronization TTL pulses were delivered to the clinical EEG acquisition system using a data acquisition board (USB-1208FS, Measurement Computing) to mark task events, including participant responses. Odors were delivered either manually using squeeze-bottles or automatically using a 12-channel computer-controlled olfactometer. Seven of the participants performed an odor detection task in which they were presented with either odorized or odorless air following a visual cue and were asked to indicate whether an odor was present (S1 Fig). Six participants performed an odor identification task in which they were presented with an odor following a verbal cue and were asked to indicate whether the odor matched the cue. In all cases, trials were separated by approximately 15 to 20 s of natural, unattended breathing of odorless air. Importantly, during these intertrial periods, participants did not anticipate receiving an odor, and no odor was ever delivered during this time. Our analyses were focused on the period occurring immediately prior to inhale onset, during a natural pause in breathing, and thus prior to any potentially confounding effects of sniffing or odor. Notably, all analyses were performed at both the group and the individual level.

**Table 1. Demographic and clinical characteristics of patients.**

| Patient | Gender | Age (years) | Age of seizure onset (years) | Epileptogenic zone | Medial temporal lobe coverage | Brain MRI | Site |
|---------|--------|-------------|------------------------------|--------------------|-------------------------------|-----------|------|
| 1 | Female | 61 | 35 | Left temporo-occipital lobe | Right and left | Remote prior left posterior parietal craniotomy, with cystic encephalomalacia and gliosis; gliosis in left parahippocampal gyrus and left middle/inferior temporal white matter | NU |
| 2 | Female | 46 | 16 | Right mesial temporal lobe | Right | Right mesial temporal sclerosis | UC |
| 3 | Female | 23 | 20 | Right temporo-parietal lobe | Right | Nodular foci (2) along lateral right temporal horn periventricular white matter | NU |
| 4 | Female | 27 | 13 | Temporal lobe | Right | Small nonspecific foci of hyperintense T2FLAIR in bilateral frontal lobe white matter and left parietal white matter | NU |
| 5 | Female | 28 | 20 | Right fronto-temporal lobe | Right | Small cavernous malformation in right medial orbital gyrus and left superior temporal gyrus; meningioma in anterior interhemispheric falx and focal cortical enhancement in left temporo-occipital region and pons | NU |
| 6 | Female | 28 | 13 | Left temporal lobe | Right and left | Normal | SU |
| 7 | Male | 20 | 15 | Left mesial temporal lobe | Right and left | Normal | UC |
| 8 | Male | 34 | 28 | Right mesial temporal lobe | Right | Normal | NU |
| 9 | Male | 32 | 22 | Left basal temporal lobe | Left | Normal | NU |
| 10 | Female | 27 | 22 | Left mesial temporal lobe | Left | Left mesial temporal sclerosis | NU |
| 11 | Female | 29 | 22 | Left temporal lobe | Left | Normal | NU |
| 12 | Female | 36 | <1 | Left mesial temporal | Left | Left mesial temporal sclerosis | NU |
| 13 | Female | 25 | 3 | Left mesial temporal | Left | Normal | NU |

**Abbreviations:** NU, Northwestern University; SU, Stanford University; UC, University of Chicago

## Respiratory and iEEG recordings

iEEG data were acquired using the clinical 128-channel EEG recording system (Nihon Kohden, Tokyo, Japan) in place at Northwestern Memorial Hospital and Stanford University Medical Center, and a clinical 256-channel EEG recording system (Natus Medical Incorporated, Pleasanton, CA) at University of Chicago Medical Center. The sampling rate for each participant was determined clinically and ranged from 500 Hz to 2000 Hz across participants. The reference and ground consisted of surgically implanted electrode strips on the surface of the brain facing towards the scalp. Depth macro-electrodes used at Northwestern University and University of Chicago were manufactured by Integra (Plainsboro, New Jersey), and depth macro-electrodes used at Sandford University were manufactured by Ad-Tech (Racine, Wisconsin).

Respiratory data were recorded using a piezoelectric pressure transducer (Salter Labs Model #5500) attached to a nasal cannula. These data were recorded by the clinical acquisition system and were therefore automatically synchronized with the iEEG data. Of note, the data acquisition system imposes a hardware high-pass filter of 0.08 Hz.

## Electrode localization

Depth electrodes were localized using pre-operative structural magnetic resonance imaging (MRI) scans and postoperative computed tomography (CT) scans using FSL's registration tool

*flirt* (FMRIB Software Library, RRID: SCR_002823) [150,151]. Individual CT images were registered to MRI images using a degree of freedom of 6, with a cost function of mutual information, which was followed by a 12 degree-of-freedom affine registration. Individual MRI images were registered to a Montreal Neurological Institute (MNI) standard brain (MNI152_1mm_-brain) with a degree of freedom of 12. The transformation matrices generated above were combined to create a transformation from the individual CT image to standard MNI space. The electrodes were localized by thresholding the raw CT image and calculating the unweighted mass center of each electrode. Finally, the coordinates were converted to standard MNI space using the transformation matrix generated above (Fig 1A).

### Phase resetting analysis (ITPC)

We used ITPC to estimate phase resetting in piriform cortex. ITPC provides an estimate of the consistency of the instantaneous phase across trials, aligned to an event, and ranges in value from 0 to 1, where 0 indicates no phase clustering, and 1 indicates perfect phase clustering [97].

### Combined ITPC analysis

iEEG data were down-sampled to 500 Hz and re-referenced to a common average. In preparation for the combined analysis, iEEG time series were normalized for each subject by subtracting the average of the time series and dividing by the standard deviation. The resulting normalized time series were then band-pass filtered at 50 log-spaced frequencies between 0.5 Hz and 200 Hz (Fig 2A), with the bandwidth logarithmically increasing from 1 Hz to 2 Hz. The signal was filtered using a two-pass, zero-phase finite impulse response filter as implemented by *fieldtrip* (RRID: SCR_004849). Subsequently, the Hilbert transform was applied to the band-pass filtered signal in order to extract the instantaneous phase of each time point and each frequency bin. The resulting phase time-series was segmented into inhale-onset-aligned epochs going from 1 s prior to, to 3 s following, the onset of the inhale. This was performed for all inhales over the course of the experiments, thus encompassing both anticipatory and nonanticipatory inhales. Because there were a larger number of nonanticipatory inhales between trials than anticipatory inhales during trials, we analyzed a subset of nonanticipatory breaths consisting of the same number as that of anticipatory trials performed for each subject. This was achieved by selecting a random subset of all nonanticipatory trials for each subject. The resulting trials were pooled over all subjects, resulting in 891 anticipatory trials and 891 nonanticipatory trials (Fig 2). The difference in ITPC value was calculated across trials at each time-frequency point, for each condition separately, using the Phase-Locking-Value index across trials [97]. The statistical significance of the ITPC for each condition was evaluated with the Rayleigh test. Multiple comparisons were corrected using the FDR method on the *P* values obtained from the Rayleigh tests.

### Individual participant ITPC analysis

To compute the ITPC for each individual participant, we followed the same steps as described above for the combined analysis, using the trials for that individual only. We then computed the maximal ITPC for each individual in the frequency range from 0.5 Hz to 2 Hz, and the time range from −1 s to 0 s (Fig 2C), or −1 s to −0.5 s and −0.5 s to 0 s (Fig 2D), for each condition. Statistical significance was determined using ANOVAs and paired *t* tests across participants.

## Pre-inhale power analyses

Power Spectral Density: PSD was computed using Matlab's *pwelch* function over the frequency range from 0 Hz to 200 Hz, with a step size of 0.05 Hz. This was computed over the entire pre-inhale time window from −1 s to 0 s. PSD was calculated separately for each participant and for each condition. The resulting PSD values were then converted to decibels, averaged across participants, and plotted, along with the standard error, across subjects (Fig 4A). Statistics were computed using paired *t* tests and FDR correction for multiple comparisons.

Time-frequency analysis of pre-inhale power: For each participant separately, iEEG data were filtered at 50 log-spaced frequencies between 0.5 Hz and 200 Hz, with the bandwidth logarithmically increasing from 1 Hz to 2 Hz. The signal was filtered using a two-pass, zero-phase finite impulse response filter as implemented by *fieldtrip*. The Hilbert transform was then applied to the band-pass-filtered signal in order to extract the instantaneous amplitude of each time point and each frequency bin. The resulting amplitude-time series were then temporally smoothed with a moving average filter kernel of 10 ms. To evaluate event-related power changes, the amplitude-time series were segmented into inhale-onset-aligned epochs going from −1 s to 0 s (at inhale onset) for each condition. Epochs were converted to power by squaring the amplitude values. The power spectrogram of each condition was separately calculated by averaging the corresponding power epochs across trials at each frequency. Then, decibel transformation was applied to the averaged spectrograms. Importantly, for this analysis, no baseline correction was performed. Raw power was used in order to avoid potentially confounding effects of baseline correction: baseline correction modifies signal-to-noise ratios, which can impact the accuracy of phase estimates [99]. This resulted in spectrograms of anticipatory and nonanticipatory conditions in the pre-inhale time window for each subject. The resulting individual-participant spectrograms were then averaged across participants. Statistics were performed using paired *t* tests across participants and FDR correction for multiple comparisons (Fig 4B, left). To confirm these findings at the individual level, we also show the maximal power for each participant across 0.5 Hz to 2 Hz and −1 s to 0 s (inhale onset) for each condition and computed a paired *t* test across participants (Fig 4B, right).

To estimate the correlation between pre-inhale ITPC and pre-inhale power, we computed the difference in maximal power across conditions for each participant and the difference in maximal ITPC across conditions for each participant. The Spearman r value was computed using Matlab's *corrcoef* function. Pearson correlation values were also computed, with no difference in results. This analysis was also conducted over a broader frequency range, 0.5 Hz to 8 Hz, with no difference in results (Fig 4C).

## Postinhale power analyses

iEEG data were preprocessed and band-pass filtered as described earlier. The Hilbert transform was applied to the band-pass-filtered signal in order to extract the instantaneous amplitude of each time point and each frequency bin. The resulting amplitude-time series were then temporally smoothed with a moving average filter kernel of 10 ms.

To evaluate event-related power changes (Fig 5A), the amplitude-time series were segmented into sniff-onset-aligned epochs going from 1 s prior to, to 3 s following, the inhale, for all anticipatory and nonanticipatory trials for each subject. All sniff-aligned epochs were converted to power by squaring the values. The power spectrogram of each condition was separately calculated by averaging the corresponding power epochs across trials at each frequency. Then, decibel transformation was applied to the averaged spectrograms, which was further normalized by subtracting a common baseline average. The baseline was defined as the time window of one second prior to the inhale onset. Finally, the power difference between

anticipatory and nonanticipatory conditions was calculated by subtracting the baseline-corrected nonanticipatory power from the baseline-corrected anticipatory power.

The statistical significance of the power change for each condition was determined using nonparametric permutation testing [152]. Surrogate events were generated by shifting actual events by a random amount. The average power across these permuted events was calculated for each permutation. A distribution of surrogate values was obtained by repeating this procedure 1,000 times. A z-score map of the actual spectrogram and its corresponding *P* values was obtained by subtracting the average of the surrogate data and dividing the result by the standard deviation of the surrogate distribution. Multiple comparisons were corrected using the FDR method on the *P* values obtained.

To perform statistics on the power difference between anticipatory and nonanticipatory conditions, we used a permutation method, in which condition labels were shuffled, and the power difference computed, at each iteration. This procedure was repeated 1,000 times to build a distribution of surrogate values of power differences at each time-frequency point across conditions. A z-score map of the actual power differences and their corresponding *P* values was obtained by subtracting the average of the surrogate data and dividing the result by the standard deviation of the surrogate distribution. Subsequently, multiple comparisons were corrected using the FDR method.

We also computed the postinhale power for each condition at the individual-participant level, using a paired *t* test for statistical comparison. For each participant, we computed the maximal power over 0.5 Hz to 12 Hz and over the time range from 0 s to 1 s postinhale. The maximal postinhale power occurred within this time and frequency window for all participants. This was done for each participant, for each condition, and statistical comparisons were conducted using a paired *t* test across participants (Fig 5B).

## Linking phase resetting and olfactory perceptual accuracy

We performed two analyses to explore the impact of phase resetting on olfactory perceptual accuracy. Given the relatively small number of incorrect trials per subject and the fact that ITPC measures stabilize at around 20 to 30 trials [98], we made use of a single-trial measure of ITPC—the DMP [105,106]. DMP is a single-trial measure of the extent of phase reset relative to the mean phase of the ITPC. Our two analyses involved first grouping trials according to their accuracy (correct or incorrect) and then computing the DMP of each group; and, second, grouping trials according to their DMP (small or large) and then computing the accuracy of each group. DMP was computed according to methods in the work by Hanslmayr and colleagues [105]. The ITPC across correct and incorrect trials was calculated for each participant. To compute the DMP of each trial for each participant, we first determined the frequency at which the maximal ITPC occurred at inhale onset. Next, the distance between the phase angle of each trial and the mean phase angle was calculated at onset (t = 0 s) using *circ_dist* as implemented in CircStat Matlab toolbox (RRID: SCR_016651). After computing phase distances for each trial for each participant, values were combined across subjects. For both analyses, bootstrapping was performed with 1,000 repetitions.

In the first analysis, for each subject whose performance on the task included both correct and incorrect trials, we grouped trials according to their accuracy (correct versus incorrect). Trials were then grouped across all subjects into two vectors (correct and incorrect). DMP values were determined using individual subject ITPC phase angles (at time 0 s and the frequency at which each individual's ITPC was maximal), and the resulting DMP values were combined across subjects for each group (correct and incorrect). This resulted in a list of DMP values for each trial, for each group (correct and incorrect). For each bootstrap repetition (*N* = 1,000), a

subset of trials from each group was randomly selected using the *randi* function in Matlab (with replacement), and the average DMP was computed for both subsets. The difference between the DMP average for correct trials and that for incorrect trials was calculated. This yielded a distribution ($N$ = 1,000) of the difference between DMP values for trials that had been grouped solely by their accuracy, with no a priori information about their DMP values (Fig 5D). Statistics were computed both under no assumption of a normal distribution and under assumption of a normal distribution. Specifically, making no assumption about the characteristics of the distribution, we computed the percentage of values in the distribution that fell above zero and computed the $P$ value from that [98]. Assuming a normal distribution, we computed z-score from the mean and standard deviation of the distribution and converted those into $P$ values [98]. Finally, we also computed the $P$ value using Matlab's *signtest* and a paired *t* test.

In the second analysis, we did the opposite: That is, we grouped trials initially by DMP values and then calculated the accuracy of each group. Trials were randomly selected as described in the first analysis above. On each bootstrap repetition, for each subject, the DMP was calculated for each trial as described above. Values were combined across participants and sorted from small to large. Then, we separated the DMP values into well-aligned and poorly aligned groups by taking the smallest 20% for the well-aligned group (closest to the mean phase) and the largest 20% for the poorly aligned group (farthest from the mean phase) (Fig 5E). The performance in each group was then calculated as percentage of correct trials over all trials. Statistics were conducted exactly as described for the first analysis above.

### Linking phase reset and olfactory motor behavior

To analyze the respiratory signals for each subject, we first normalized the data by subtracting the mean and dividing by the standard deviation of the entire respiratory signal (Fig 6). For each subject, inhale durations were computed as the time between inhale onset and the first zero-crossing. The peaks were computed as the maximum airflow value over the duration of inhale. To perform analysis on the respiratory data at the individual level, we used the DMP measure to estimate single-trial phase resets (using the same procedure and parameters as described above) and then determined whether these phase resets related to inhale sizes. DMP was computed for each trial for each participant. We then defined well-aligned and poorly aligned trials using a median split and computed inhale peaks for each trial type. Statistics were performed using a paired *t* test across participants.

### Supporting information

**S1 Fig. Experimental paradigms.** Related to Fig 1. Following selection of data sets according to our inclusion criteria (described in detail in the main text), data from two olfactory tasks were included. In a detection task (top panel), the participants were presented with either odorized or odorless air following a visual cue and indicated whether an odor was present via button press. In an identification task (bottom panel), the participants were presented with an odor following an auditory cue and indicated whether the odor matched the cue via button press. The red outlined area (box just prior to inhale onset) indicates the time window of interest in our analyses, specifically, the pre-inhale anticipatory period.
(TIF)

### Acknowledgments

We thank Navid Shadlou, Enelsa Lopez and Jeremy Eagles for assistance with data collection.

## Author Contributions

**Conceptualization:** Ghazaleh Arabkheradmand, Guangyu Zhou, Torben Noto, Qiaohan Yang, Gregory Lane, Christina Zelano.

**Data curation:** Gregory Lane, Christina Zelano.

**Formal analysis:** Ghazaleh Arabkheradmand, Guangyu Zhou, Christina Zelano.

**Funding acquisition:** Christina Zelano.

**Methodology:** Ghazaleh Arabkheradmand, Guangyu Zhou, Joshua M. Rosenow, Gregory Lane, Christina Zelano.

**Resources:** Stephan U. Schuele, Josef Parvizi, Jay A. Gottfried, Shasha Wu, Joshua M. Rosenow, Mohamad Z. Koubeissi.

**Software:** Ghazaleh Arabkheradmand, Guangyu Zhou.

**Supervision:** Guangyu Zhou, Gregory Lane, Christina Zelano.

**Visualization:** Ghazaleh Arabkheradmand, Guangyu Zhou, Gregory Lane, Christina Zelano.

**Writing – original draft:** Ghazaleh Arabkheradmand, Guangyu Zhou, Gregory Lane, Christina Zelano.

**Writing – review & editing:** Guangyu Zhou, Torben Noto, Qiaohan Yang, Jay A. Gottfried, Gregory Lane, Christina Zelano.

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
