## [Editor Report · Decision Letter 0]

26 Feb 2020

Dear Dr Zhou, 

Thank you for submitting your manuscript entitled "Anticipation-induced delta phase reset improves human olfactory perception" for consideration as a Research Article by PLOS Biology.

Your manuscript has now been evaluated by the PLOS Biology editorial staff, as well as by an Academic editor with relevant expertise, and I am writing to let you know that we would like to send your submission out for external peer review. Please accept my apologies for the delay in sending this initial decision to you.

Before we can send your manuscript to reviewers, we need you to complete your submission by providing the metadata that is required for full assessment. To this end, please login to Editorial Manager where you will find the paper in the 'Submissions Needing Revisions' folder on your homepage. Please click 'Revise Submission' from the Action Links and complete all additional questions in the submission questionnaire.

Please re-submit your manuscript within two working days, i.e. by Feb 28 2020 11:59PM.

Kind regards,

Gabriel Gasque, Ph.D.,

Senior Editor

PLOS Biology

---

## [Decision Letter · Decision Letter 1]

25 Mar 2020

Dear Dr Zhou,

Thank you very much for submitting your manuscript "Anticipation-induced delta phase reset improves human olfactory perception" for consideration as a Research Article by PLOS Biology. As with all papers reviewed by the journal, yours was evaluated by the PLOS Biology editors as well as by an Academic Editor with relevant expertise and by thee independent reviewers. You will note that reviewer 3, Nathan Weisz, has signed his comments. 

Based on the reviews, we are positive about your manuscript. However, before we can make any decision about publication, you need to revise your manuscript to address the points raised by the reviewers; particularly reviewer 3's comment about whether the anticipatory effect is not due to smearing of the time-frequency analysis. 

Please also make sure to address the data and other policy-related requests noted at the end of this email.

We expect to receive your revised manuscript within two weeks. 

In addition to the remaining revisions and before we will be able to formally accept your manuscript and consider it "in press", we also need to ensure that your article conforms to our guidelines. A member of our team will be in touch shortly with a set of requests. As we can't proceed until these requirements are met, your swift response will help prevent delays to publication.

*Copyediting*

*Published Peer Review History*

*Early Version*

*Submitting Your Revision*

Sincerely,

Gabriel Gasque, Ph.D., 

Senior Editor

PLOS Biology

ETHICS STATEMENT:

--Please indicate within your manuscript if the protocols approved by Institutional Review Boards of Northwestern University, University of Chicago and Stanford University adhered to the declaration of Helsinki or any other national or international ethical guidelines. 

DATA POLICY:

Note that we do not require all raw data. Rather, we ask for all individual quantitative observations that underlie the data summarized in the figures and results of your paper. For an example see here: http://www.plosbiology.org/article/info%3Adoi%2F10.1371%2Fjournal.pbio.1001908#s5

These data can be made available in one of the following forms:

Regardless of the method selected, please ensure that you provide the individual numerical values that underlie the summary data displayed in the following figure: Figures 2C-E, 3B, 4A-C, 5B-E, and 6A-D.

Please also ensure that figure legends in your manuscript include information on where the underlying data can be found and that your supplemental data file/s has a legend.

Reviewer remarks:

Reviewer #1: This is an outstanding and well written paper describing possible neural mechanisms underlying anticipation's effect on odor perception -- namely the well established influence of anticipation/attention on odor perception. The results are remarkably well presented, methods carefully described and the introduction and discussion well crafted with a crisp scholarly tone. While this manuscript is human iEEG, the authors do service to this study by placing their results in context with those from the non-human literature. Compelling iEEG of this sorts is exceptionally rare in the olfactory system and thus this project has a high level of innovation and the outcomes, that delta phase resetting independent of sensori-motor influences, may sculpt perception, is significant. This work will be of impact to the olfactory community but more than that, to the wealth of researchers exploring the brain basis for attention.

We have no major issues, nor any true minor issues for that matter -- clearly this manuscript has been polished by the authors over rounds of careful consideration. 

Small items for correction:

1)Line 483: What exactly is meant by power modulations of higher Hz (perhaps cite a fig here to guide readers)?

2)Fig 1: one of the pink shaded boxes overshadows the raw LFP trace.

3)The last paragraph in the Discussion on neuroD disorders, etc, is super tangential. We get it, but, wouldn't this paper end more compelling with highlighting really why IT is great (as well covered in the previous paragraphs) versus needing to end with a weak translational spin-off/foreshadow? 

Reviewer #2: The study by Ghazaleh et al. describes how the phase of low frequency local field potential (LFP) oscillations recorded by intracranial EEG in humans are modulated in the piriform cortex by the anticipation of an odor prior to sniffing. The study is important, as most research thus far targeting the mechanisms of attention was focused on the auditory and visual systems, which have markedly different anatomy. Therefore, it is not clear if the same principles can be applied to the olfactory system. Ghazaleh et al. provide convincing evidence that phase reset, which is a major mechanism of attention in other sensory systems, is also engaged by olfactory perceptual processes. I reviewed a previous version of the manuscript for a different journal, and I find this version much improved. In fact, I only have a couple of minor criticisms/suggestions.

1) I have a hunch that the "odor-evoked theta power increase" in the olfactory anticipation condition might be explained by the consistent delta phase reset and phase amplitude coupling. Even if the authors disagree, this possibility should be mentioned and disputed. 

2) In the results, the frequency range 0-2 Hz or 0-8 Hz does not really make sense, as there were no measurements made at 0 Hz. Therefore, I suggest changing it to 0.5 - xxx Hz everywhere, as based on the methods, 0.5 Hz was the low end of the frequency spectrum tested.

Reviewer #3, Nathan Weisz: This is an interesting paper investigating time-resolved neuronal dynamics in olfactory (piriform) cortical strutures, during a pre-inhalation period when human subjects either anticipated or did not anticipate an odour presentation. Measurements were done in the context of presurgical epilepsy diagnosis using sEEG directly from relevant cortical structures. The authors show in particular increased intertrial phase locking in a delta frequency range prior to inhalation when individuals were anticipating an odour. This result was correlated with post-inhalation power modulations as well as behavioural performance. The authors show that the effect is not correlated with post-stimulus power changes as well as overt inhalation responses. Overall, the study points to similar anticipation / attention (not dissociable given the current design) related phase adjustment in a slow frequency range, similar as it has been shown in other sensory modalities. Given the unique architecture of the olfactory system, this is an interesting study that is a strong candidate for Plos Biology.

For me the most critical issue is whether the main effect is indeed a unique anticipation related pre-inhalation effect. I do not find this completely conclusive given the presentation of the results. In particular it can be seen in Figure 2A that the peak (at ~.7 Hz) is clearly post-stimulus, without an obvious distinct prestimulus peak. Depending on the settings of the time-frequency analysis, the pre-inhalation effects (in particular at this low frequency) could be a "bleeding effect" of a post-inhalation peak into the prestimulus period. I would suggest rerunning the time-frequency analysis using an FFT on hanning-tapered windows of varying lengths. If the pre-inhalation effect shrinks in the post-inhalation direction, then I would be worried about the current interpretation. Furthermore it would be also a good idea to present the broadband evoked responses: is a low frequency difference visible before spectral analysis? is there a post-stimulus difference in terms of latencies (e.g. an earlier peak when anticipating an odour).

The authors report an absence of power differences. This by itself is not shocking, however the spectra that are presented in Figure 4 look very "1/f"-ish without any clear peaks. Wouldn't this be expected when recording from medial temporal lobe structures (e.g. a theta peak). Also, it would be a good idea to display the t-map in 4B using a more meaningful minimum / maximum (the color bar says -30 to +30).

In their discussion the authors mention fMRI studies showing anticipation related differences (e.g. p. 22 ll. 441). In the present study effects are only found in terms of intertrial phase coherence. How can these effects be reconciled?

Minor:

-p. 5, l. 97:  "enhances olfactory response ..."  "improves"?

-In my personal opinion the long excursion into Predictive Coding on p. 23 can be removed without much loss in substance.

---

## [Editor Report · Decision Letter 2]

24 Apr 2020

Dear Dr Zhou,

On behalf of my colleagues and the Academic Editor, Simon Hanslmayr, I am pleased to inform you that we will be delighted to publish your Research Article in PLOS Biology. 

Early Version

PRESS 

Kind regards,

Vita Usova 

Publication Assistant, 

PLOS Biology

on behalf of

Gabriel Gasque,

Senior Editor

PLOS Biology